# Pragmatic accuracy of an in-house loop-mediated isothermal amplification (LAMP) for diagnosis of pulmonary tuberculosis in a Thai community hospital

Sarawut Toonkomdang[1], Phichayut Phinyo[2,3]*, Benjawan Phetsuksiri[4], Jayanton Patumanond[3], Janisara Rudeeaneksin[4]ꞏ, Wiphat Klayut[4]ꞏ

1 Department of Medical Technology, Maesot General Hospital, Tak, Thailand, 2 Department of Family Medicine, Faculty of Medicine, Chiang Mai University, Chiang Mai, Thailand, 3 Center for Clinical Epidemiology and Clinical Statistics, Faculty of Medicine, Chiang Mai University, Chiang Mai, Thailand, 4 Department of Medical Sciences, National Institute of Health, Ministry of Public Health, Nonthaburi, Thailand

ꞏ These authors contributed equally to this work.
* phichayutphinyo@gmail.com

**Data Availability Statement:** All relevant data are within the manuscript and its Supporting Information files.

## Abstract

### Background

To improve the quality of diagnosing pulmonary tuberculosis (TB), WHO recommends the use of rapid molecular testing as an alternative to conventional microscopic methods. Loop-mediated isothermal amplification assay (LAMP test) is a practical and cost-effective nucleic amplification technique. We evaluated the pragmatic accuracy of an in-house LAMP assay for the diagnosis of TB in a remote health care setting where an advanced rapid molecular test is not available.

### Methods

A prospective diagnostic accuracy study was conducted. Patients with clinical symptoms suggestive of TB were consecutively enrolled from April to August 2016. Sputum samples were collected from each patient and were sent for microscopic examination (both acid-fast stain and fluorescence stain), in-house LAMP test, and TB culture.

### Results

One hundred and seven patients with TB symptoms were used in the final analysis. This included 50 (46.7%) culture-positive TB patients and 57 (53.3%) culture-negative patients. The overall sensitivity of the in-house LAMP based on culture positivity was 88.8% (95/107) with a 95%CI of 81.2–94.1. The sensitivity was 90.9% (40/44) with a 95%CI of 78.3–97.5 for smear-positive, culture-positive patients, and was 16.7% (1/6) with a 95%CI of 0.4–64.1 for smear-negative, culture-positive patients. The overall sensitivity of the in-house LAMP test compared to smear microscopy methods were not significantly different (p = 0.375). The

**Funding:** The author(s) received no specific funding for this work.

**Competing interests:** The authors have declared that no competing interests exist.

specificity of the in-house LAMP based on non-TB patients (smear-negative, culture-negative) was 94.7% (54/57) with a 95%CI of 85.4–98.9.

## Conclusions

The diagnostic accuracy of the in-house LAMP test in a community hospital was comparable to other previous reports in terms of specificity. The sensitivity of the in-house assay could be improved with better sputum processing and DNA extraction method.

## Introduction

Tuberculosis (TB), an airborne communicable disease, has long been considered a significant threat to global public health. According to The World Health Organization (WHO), 10 million people were newly infected with TB in 2018 [1]. The incidence and prevalence of TB vary greatly across the globe, 87% of total cases resided within 30 countries with a high TB burden. In Thailand, the incidence rate was 153 cases per 100,000 population in 2018. Early diagnosis and timely treatment is an essential component of The End TB Strategy endorsed by the WHO, aiming to end the global TB epidemic by the year 2035 [2]. However, TB is still under-diagnosed and undertreated, especially in resource-limited countries, due to the lack of highly sensitive and specific diagnostic tools which are usually expensive and require adequate infrastructure [1,3]. Novel diagnostic methods with enough simplicity and cost-effectiveness are therefore necessary to improve the accurate identification of TB patients in those resource-limited settings [3,4].

Molecular testing methods such as polymerase chain reaction (PCR) or Xpert MTB/RIF have been widely acknowledged as alternative tools to TB culture for the diagnosis of TB patients [3,5]. These nucleic amplification techniques were known for yielding rapid and accurate TB diagnosis, which would overcome the limitations of classical methods (ie, insensitivity for smear microscopy and lengthy incubation period for TB culture). However, several obstacles remain for the application of these molecular tests as point-of-care testing in community settings due to their complexity to execute and substantial requirements for financial and personnel resources [3,6]. Loop-mediated isothermal amplification (LAMP) assay is another recently developed nucleic acid amplification technique. Unlike PCR, where the amplification of DNA fragment occurs in temperature-dependent steps, the reaction of LAMP assay functions in isothermal or constant temperature conditions [7,8]. In 2016, WHO endorsed the use of commercial TB-LAMP assay (Eiken Chemical Co., Tokyo, Japan) as a replacement for smear microscopy for the diagnosis of TB [9]. TB-LAMP assay has a low cost per test, does not required advanced technological facilities, and can be routinely practiced in general hospital laboratories [6,10].

As financial resources are usually limited in countries with high TB prevalence, a commercial TB-LAMP could still be unattainable. An in-house LAMP assay may be more affordable to the commercial one [11–15]. However, it did require extra-training and skill of technicians to process the clinical specimens. In the past decades, several clinical studies and meta-analyses had evaluated the diagnostic accuracy of in-house LAMP tests for the diagnosis of pulmonary TB [14,16,17] (S1 Table). From the latest meta-analysis, the overall sensitivity and specificity of in-house LAMP tests was 93.0% (95%CI 88.9–95.7) and 91.8% (95%CI 86.4–95.1), respectively [17]. One recent study in Thailand reported the sensitivity and the specificity of the in-house LAMP at 94.4% (95%CI 88.9–97.7) and 94.3% (95%CI 87.2–98.1), respectively [15]. However,

the reported accuracy could be overestimated if it is assessed in qualified laboratories with highly skilled technicians and sufficient resources where molecular tests are usually available [17]. Therefore, this study aimed to evaluate the pragmatic accuracy of the in-house LAMP assay for the diagnosis of pulmonary TB in a community hospital of a developing country with a high TB burden.

## Materials and methods

### Ethics statement

This study was approved by the Research Ethics Committee of Maesot General Hospital, The Ministry of Public Health (serial number 37/2015) and The Human Research Ethics Committee of Thammasat University, Faculty of Medicine (COA number 081/2016). The clinical samples used in this study were collected from all patients as routinely done. Informed consent was obtained from all patients prior to inclusion.

### Setting

The study was performed in Maesot General Hospital, a large-sized community hospital with 365 in-patient beds. The hospital is located in Maesot District, Tak Province, which shares the border with Myanmar. The hospital provides standard health care to both Thai and non-Thai patients (Burmese immigrants and ethnic minorities). According to the Health Data Center, the Ministry of Public Health in Thailand, the incidence rate of pulmonary TB in Maesot was 351 per 100,000 in 2019. The level of the health care system of the hospital is considered rural. Maesot hospital has a reference laboratory with biosafety cabinet infrastructure, BSC class II. There are four lab technicians and one lab assistant within each working shift. Power generator (350 kW) and UPS (2.7 kW) were available in case of power outages, which was infrequent. The median LAMP test workload per day was 6 (range 4–10).

### Study design

This prospective diagnostic accuracy research was conducted from April to August 2016. Adult patients aged more than 15 years old with symptoms indicative of pulmonary TB (coughing for more than two weeks with or without hemoptysis) and no history of TB were consecutively enrolled regardless of nationality status. Patients with previously documented TB history or patients with two contaminated or missing cultures were excluded from the study.

### Methods

All patients were given three sealed containers for the collection of morning sputum specimens. Only one sputum specimen with adequate sputum containing both mucoid or mucopurulent characters and a sample volume of more than 3 ml was selected to be used in all investigation procedures. Specimens were sent for smear microscopy with conventional acid-fast bacilli (AFB) staining with Ziehl-Neelsen technique and fluorescence acid-fast staining with Auramine O solution. The smear-positive case was defined according to WHO definitions as the presence of at least two smears of scanty grade, or one or more smears of 1+ or more. A smear negative case was conversely defined. All patients with symptoms suggestive of TB were offered routine HIV counselling and HIV rapid antibody tests.

**Sputum decontamination and culture examination.** For the sputum decontamination process, the collected samples and 2% N-Acetyl-L-cysteine (NALC) NaOH were poured into a 50 ml sterile centrifuge tube in an equal proportion. The specimens were subsequently mixed

by vortexing for 30 seconds and left at room temperature (20–25˚C) for 15 minutes. Then, the test tubes were filled with phosphate buffer saline (pH 6.8) until the volume reached the level of 50 ml. The samples were put in a high-speed refrigerated centrifuge at 3,000 g for 20 minutes. Next, the supernatants were poured off, leaving the tube with decontaminated sputum samples. Finally, a drop (1 ml) of phosphate buffer saline (pH 6.8) was used for resuspension of the specimens.

For TB culture, the reference test, we performed both conventional culture method on L-J (Lowenstein-Jensen) medium and BBL MGIT 960 (mycobacterial growth indicator tube) culture method. The culture media were inoculated with processed sputum specimens and incubated at 35 to 37˚C and monitored weekly for growth until 8 weeks. The sputum samples were considered as "culture-positive" if growth was detected in either of L-J or MGIT culture, regardless of the smear status. If growth was not detected in neither of the culture methods and both microscopy results were negative, the samples were considered as "culture-negative" or "non-TB patients". Patients with smear-positive and culture-negative, which were generally considered as probable TB, were excluded from the analysis. Both smear microscopy and culture methods were performed according to the standard protocols [18].

**In-house LAMP test.** The LAMP test consists of three steps as follows: DNA extraction, isothermal amplification, and visual interpretation with fluorescence. In this study, we followed the TB Fast AMP technique, which was developed by the National Institute of Health of Thailand and was described in our previous studies [13,15,19]. The procedures were described as follow. Flexi Gene® DNA Kit (Qiagen co., USA) and Protenase K Kit (Qiagen co., USA) were used for DNA extraction. Six primers were used for the recognition of eight distinct regions on the 16S ribosomal RNA gene of *Mycobacterium tuberculosis*. Each single LAMP reaction includes 12 μl of TB-Fast AMP mixture (FastAMP master mix includes 2 μl 10Xbuffer, 4 μl 2mM dNTPs, 3.2 μl 5M betaine, 1.2 μl 100 mM $MgSO_4$, 1.6 μl primer mixture), 1 μl *Bst* DNA polymerase enzyme (New England Biolabs, Ipswich MA, USA), 1 μl fluorescent detection reagent (FDR; Eiken Chemical Tokyo, Japan) and 6 μl of extracted DNA samples. Amplification of reaction mixture was performed in the heating blocks at 65˚C for 60 minutes, then examined directly by visual observation. The LAMP assay was considered "positive" if the color of the reaction mixture changed from orange to green, or fluorescence was directly observed with the naked eyes. The test was considered "negative" if the color of the mixture remained unchanged. For quality control, positive control (test tube with *M. tuberculosis* genetic materials) and negative control (test tube without *M. tuberculosis* genetic materials) were included in all runs.

## Statistical analysis

We used Fisher's exact probability test for comparison of differences in independent proportions and Student's t-test for two independent means. The sensitivity, specificity, positive predictive values (PPV), negative predictive values (NPV), and positive and negative likelihood ratios were calculated and reported with its 95% confidence interval. The 95% confidence interval were estimated using the Clopper Pearson binomial exact method. The comparison of sensitivity, specificity, and overall test accuracy between the LAMP test and smear microscopy methods was performed with McNemar's exact probability test. Pairwise testing to compare the specificity between the LAMP test and the smear microscopy methods was not performed as the specificity of the latter was affected by incorporation bias and would not be comparable to the in-house LAMP. The inter-rater reliability and the correlation of the LAMP test with smear microscopy methods was analyzed with Kappa's statistics and Spearman's rank correlation, respectively. P-values of less than 0.05 were considered statistically significant. All statistical analyses were done using Stata version 16 (StataCorp, Texas).

## Results

A total of 120 patients to be evaluated for TB were consecutively included from April to August 2016. Three patients with two contaminated cultures, two patients who subsequently were detected as previously documented TB cases, and eight patients who had smear-positive and culture-negative results were excluded from the analysis; only 107 patients remained in the study (Fig 1). Most of the included patients were male (60% vs. 40%) with a mean age of 47 years. Fifty (46.7%) were culture-positive TB patients and 57 (53.3%) were culture-negative patients. The baseline demographic data between culture-positive and culture-negative patients were comparable (Table 1). For clinical characteristics, the presence of cavitary lesions on chest radiographs and the character of collected sputum was statistically different. Culture-positive TB patients had higher proportion of cavitary lesions (14.0% vs. 1.8%, p = 0.024) and mucous sputum specimen (52.0% vs 24.6%, p = 0.005) than those with negative TB culture. The proportion of patients with salivary sputum was significantly lower than mucous sputum in both smear-positive and LAMP-positive results (31.3% vs. 57.5%, p = 0.009 and 29.9% vs. 60.0%, p = 0.003, respectively). All included patients had negative HIV results.

The overall sensitivity of the LAMP test was 82.0% (95%CI 68.6–91.4), whereas the sensitivity in smear-positive, culture-positive patients and smear-negative, culture-positive was 90.9% (95%CI 78.3–97.5) and 16.7% (95%CI 0.4–64.1), respectively. The overall sensitivity of both the AFB and the fluorescence stain was slightly higher than that of the LAMP test; however, the differences were non-significant (Table 2). The specificity, positive predictive value, and negative predictive value of the LAMP test was 94.7% (95%CI 85.4–98.9), 93.2% (95%CI 81.3–98.6), and 85.7% (95%CI 74.6–93.3), respectively. The positive and negative likelihood ratios of the LAMP test was 15.6 (95%CI 4.47–82.12) and 0.19 (95%CI 0.08–0.44), respectively. The accuracy measures for the diagnosis of TB cases were shown to vary across different test methods (LAMP test, AFB stain, and fluorescence stain), the differences were without statistical significance (Table 2).

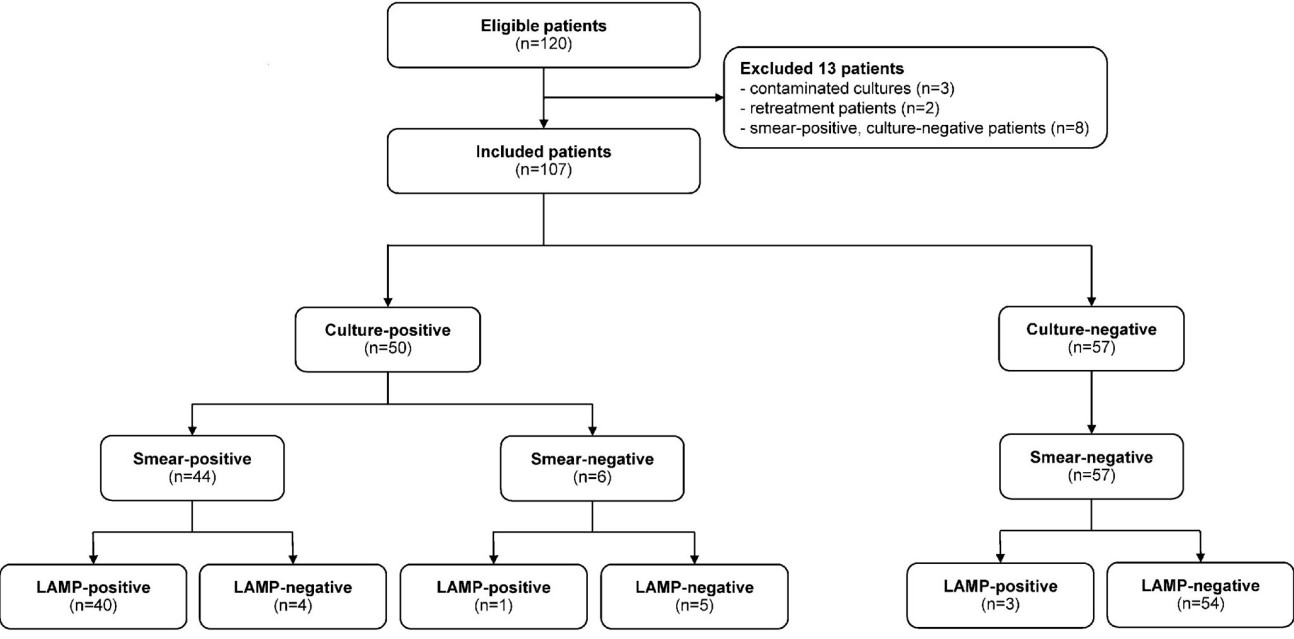

**Fig 1. Study flow diagram of patient enrollment and results of index and reference test based on culture result.**

**Table 1. Demographic and clinical characteristics of the patients by TB culture status.**

| Characteristics | TB Culture Positive (S+ or S-, C+) | TB Culture Negative (S-, C-) | P-Value |
|---|---|---|---|
| | **n = 50 (46.7%)** | **n = 57 (53.3%)** | |
| Gender | | | |
| Male | 30 (60.0) | 36 (63.2) | 0.842 |
| Female | 20 (40.0) | 21 (36.8) | |
| Nationality | | | |
| Thai | 28 (56.0) | 21 (36.8) | 0.054 |
| Non-Thai | 22 (44.0) | 36 (63.2) | |
| Age (year, mean±SD) | 48.7±17.4 | 45.8±18.7 | 0.408 |
| Chest radiographs | | | |
| Without cavitary lesions | 43 (86.0) | 56 (98.2) | 0.024 |
| With cavitary lesions | 7 (14.0) | 1 (1.8) | |
| Character of sputum | | | |
| Salivary | 24 (48.0) | 43 (75.4) | 0.005 |
| Mucous | 26 (52.0) | 14 (24.6) | |

Abbreviations: TB, tuberculosis; C, culture (+ positive or −negative); S, smear microscopy (+ positive or −negative); SD, standard deviation.

LAMP test results were highly correlated with those of AFB and fluorescence stain (Spearman's rho 0.85, p<0.001) in the diagnosis of culture-positive TB cases (Table 3). The in-house LAMP also showed substantial to an almost perfect agreement with both microscopy methods in the diagnosis of culture-positive cases (Kappa 0.85, 95%CI 0.74–0.95, p<0.001) (Table 3).

## Discussion

This study has demonstrated the pragmatic diagnostic performance of our in-house LAMP assay in a remote hospital of a high TB burden country. The overall sensitivity was lower than

**Table 2. Diagnostic accuracy of the in-house LAMP test, AFB stain, and fluorescence stain.**

| Method | Sensitivity% (95% CI), no. corrects | | | Specificity% (95% CI), no. corrects | Accuracy% (95%CI), no. corrects (n = 107) | PPV% (95% CI) | NPV (95% CI) | LR+ (95% CI) | LR- (95% CI) |
|---|---|---|---|---|---|---|---|---|---|
| | **S+, C+ (n = 44)** | **S-, C+ (n = 6)** | **Any S, C+ (n = 50)** | **S-, C- (n = 57)** | | | | | |
| LAMP | 90.9 (78.3,97.5), N = 40 | 16.7 (0.4,64.1), n = 1 | 82.0 (68.6,91.4), n = 41 | 94.7 (85.4,98.9), n = 54 | 88.8 (81.2,94.1), n = 95 | 93.2 (81.3,98.6) | 85.7 (74.6,93.3) | 15.6 (4.5,82.1) | 0.2 (0.1,0.4) |
| AFB stain | - | - | 88.0 (75.7,95.5), n = 44 | 100.0 (93.7,100.0), n = 57 | 94.4 (88.2,97.9), n = 101 | 100.0 (93.7,100.0) | 90.5 (80.4,96.4) | - | - |
| Fluorescence stain | - | - | 88.0 (75.7,95.5), n = 44 | 100.0 (93.7,100.0), n = 57 | 94.4 (88.2,97.9), n = 101 | 100.0 (93.7,100.0) | 90.5 (80.4,96.4) | - | - |
| LAMP test vs. AFB stain | | | P = 0.375* | P = 0.250* | P = 1.000* | | | | |
| LAMP test vs. Fluorescence stain | | | P = 0.375* | P = 0.250* | P = 1.000* | | | | |

*P-values from McNemar's Exact probability test.

Abbreviations: AFB, acid fast bacilli; C, culture (+ positive or −negative); CI, confidence interval; LAMP, loop-mediated isothermal amplification; LR+, positive likelihood ratio; LR-, negative likelihood ratio; no. correct, number correctly identified; NPV, negative predictive value; PPV, positive predictive value; S, smear microscopy (+ positive or −negative).

**Table 3. Inter-rater reliability and diagnostic agreement between an in-house LAMP test and AFB stain-fluorescence stain.**

| LAMP Test | AFB Stain & Fluorescence stain | | |
|---|---|---|---|
| | **Positive** | **Negative** | **Total** |
| Positive | 40 | 4 | 44 |
| Negative | 4 | 59 | 63 |
| Total | 44 | 63 | 107 |
| Agreement (%) | 92.5% | | |
| Kappa (95%CI, p-value) | 0.85 (0.74–0.95, p<0.001) | | |
| Spearman's rho (p-value) | 0.85 (p<0.001) * | | |

Abbreviations: LAMP, loop-mediated isothermal amplification; CI, confidence interval.

*95% confidence interval of Spearman's rank correlation is not estimable.

the majority of the previous in-house LAMP studies [11,15,20–23]. Nonetheless, the specificity was comparable to other figures reported in the literature [11,12,15,21,22]. In comparison to microscopy methods (AFB and fluorescence stain), the in-house LAMP was inferior in terms of overall sensitivity. Based on the result of our study, we suggest that the in-house LAMP should not be a substitute to conventional smear methods, but should be done in parallel, which would result in a higher sensitivity with fewer false-negative TB cases.

In the past, several studies reported a higher sensitivity of in-house LAMP tests, ranging from 90.0 to 100.0% [11,15,20–25]. Most of these studies were reported from either university hospitals, TB-specialized centers or hospitals, or national TB-specialized laboratories, which were generally equipped with highly trained personnel and adequate infrastructural supports [17]. The overall sensitivity of our in-house LAMP was consistent with two previous studies from India and Zambia, which was 79.5% (95%CI 64.0–89.0) and 81.4% (95% CI 71.6–89.0), respectively [12,16]. Although both studies were performed in university hospitals, the LAMP procedures were modified to suit local conditions, and sputum processing and DNA extraction were done with commercial kits. The higher sensitivity of the acid-fast stain and the fluorescence stain in our study could be explained by the high prevalence of TB, the absence of HIV patients or fewer patients with paucibacillary sputum, and the availability of skilled technicians [16,26–28]. Besides, specimen decontamination with concentrated NaOH decreases the amount of viable genetic materials for amplification, which could reduce the sensitivity of both the LAMP test and TB cultures. A lower concentration of NaOH (1–1.5%) or NaOH free methods during sample decontamination may be suggested [16,29]. The sensitivity of the LAMP test in smear-negative specimens could not be accurately estimated in this study as there were too few smear-negative, culture-positive patients.

The overall specificity of the LAMP test was 94.7% (95%CI 85.4–98.9) for non-TB patients. This was in concordance with a recent meta-analysis, which reported pooled specificity of in-house LAMP tests of 91.8% (95%CI 86.4–95.1) [17]. However, the specificity of the in-house assays was lower than that of the Loopamp commercial kit, which was reported at 96.5% (95% CI 94.7–97.7). A false positive LAMP result in smear-positive cases was frequently encountered in routine practice, which could be explained by multiple factors such as higher temperature, higher humidity, suboptimal reagents volume, and crossover contamination [17,30]. For temperature, only available water bath was applied for temperature controls during LAMP procedures instead of a more stable dry heating block. A recent study suggested a high reaction volume of 30–35 µl due to the risk of self-priming in concentrated reagents [30].

Currently, the WHO only endorses the use of two rapid molecular tests for the diagnosis of pulmonary TB, which were Xpert MTB/RIF and the commercialized TB-LAMP assay [9]. According to previous studies, both had shown comparable performance in smear-positive samples, but higher sensitivity was shown in Xpert MTB/RIF than in the LAMP test [6,12]. Xpert MTB/RIF has been endorsed for use in the diagnosis of TB in many countries, including Thailand [4,31]. However, only a portion of patients, excluding foreigners and ethnic minorities, could reimburse the cost for Xpert MTB/RIF due to the regulation stated by The National Health Security Office (NHSO). To better control the spread of TB, access to rapid diagnostic tools should be provided to all patients with symptoms suggestive of TB [3]. Thus, a LAMP assay may be more applicable in terms of accessibility and affordability, especially in the decentralized areas [4,32].

However, there were some limitations to this study. First, the study size may not be substantial enough to provide the power required to detect a statistically significant difference between tests. Second, no patients with HIV infection were included during the study period, as HIV status could be influential to the diagnostic performance of both the smear microscopy and the LAMP test, especially in areas with a high prevalence of TB-HIV coinfection. Third, there was a higher proportion of salivary sputum than mucous sputum in this study. This could affect the diagnostic performance of both the index and the reference test [33]. Both the quality and quantity of sputum specimens were associated with the positivity of smear, molecular testing methods (Xpert MTB/RIF and PCR), and TB culture [34,35]. Thus, some patients with pulmonary TB might be classified as smear-negative, LAMP-negative, or even culture-negative cases.

Finally, the use of routine TB culture as a reference standard might be inadequate, as some TB patients could be classified as not having TB [6]. With a higher quality reference standard, the sensitivity of the in-house LAMP should be increased when a portion of three remaining false-positive cases was re-classified as true-positive cases. Different culture media and techniques could be used in composite to achieve different performance characteristics [36]. In our study, two different culture techniques, L-J and MGIT, were used to increase the diagnostic rate of TB [37]. We also applied a strict diagnostic definition in calculating specificity by considering only patients with smear-negative and culture-negative results [38].

## Conclusions

In conclusion, a LAMP test is a practical and affordable nucleic amplification technique for the diagnosis of pulmonary TB, which should be implemented in resource-limited settings where Xpert MTB/RIF is unavailable. The diagnostic accuracy of the in-house LAMP was similar to previous studies for specificity. To improve the test sensitivity, a better sputum processing and DNA extraction method is essential. The in-house LAMP test had lower sensitivity than smear microscopy. Therefore, a parallel examination of both smear microscopy and the in-house LAMP test is suggested to minimize the risk of false-negative results, especially in an endemic area.

## Supporting information

**S1 Table. Review on diagnostic accuracy of in-house LAMP assays for diagnosis of pulmonary tuberculosis.**
(DOCX)

**S2 Table. LAMP minimal dataset.**
(CSV)

## Acknowledgments

The authors wish to acknowledge the contributions of all the medical and nursing staff of the TB clinic at Maesot hospital for their help in data collection, and all relevant personnel of The National Institute of Health, Department of Medical Science, The Ministry of Public Health for their technical advice and support.

## Author Contributions

**Conceptualization:** Sarawut Toonkomdang, Phichayut Phinyo.

**Data curation:** Sarawut Toonkomdang, Benjawan Phetsuksiri, Jayanton Patumanond.

**Formal analysis:** Phichayut Phinyo, Jayanton Patumanond.

**Investigation:** Sarawut Toonkomdang, Benjawan Phetsuksiri, Janisara Rudeeaneksin, Wiphat Klayut.

**Methodology:** Sarawut Toonkomdang, Phichayut Phinyo, Jayanton Patumanond.

**Project administration:** Sarawut Toonkomdang.

**Resources:** Benjawan Phetsuksiri, Janisara Rudeeaneksin, Wiphat Klayut.

**Software:** Phichayut Phinyo, Jayanton Patumanond.

**Supervision:** Phichayut Phinyo.

**Validation:** Benjawan Phetsuksiri, Janisara Rudeeaneksin, Wiphat Klayut.

**Visualization:** Phichayut Phinyo.

**Writing – original draft:** Sarawut Toonkomdang.

**Writing – review & editing:** Phichayut Phinyo, Jayanton Patumanond.

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
