## [Decision Letter · Decision Letter 0]

4 Mar 2020

PONE-D-20-00432

Pragmatic accuracy of loop-mediated isothermal amplification (LAMP) for diagnosis of pulmonary tuberculosis in a Thai community hospital

PLOS ONE

Dear Dr. Phinyo,

Thank you for submitting your manuscript to PLOS ONE. After careful consideration, we feel that it has merit but does not fully meet PLOS ONE’s publication criteria as it currently stands. Therefore, we invite you to submit a revised version of the manuscript that addresses the points raised during the review process.

I have received the reviews of your manuscript. While your paper addresses an interesting question, both reviewers expressed significant concern about your study and did not recommend publication in its present form.  Please see reviewers’ insightful comments below and address their comments carefully.  In particular, both reviewers have questions regarding the performance of the TB LAMP test performed in this study compare to the other TB LAMP tests in other published literature as well as WHO endorsed commercial LAMP test.  Both reviewers also felt that the effects and definition of a gold standard for TB testing which is not itself perfect should be discussed and reconsidered.

We would appreciate receiving your revised manuscript by Apr 18 2020 11:59PM. To enhance the reproducibility of your results, we recommend that if applicable you deposit your laboratory protocols in protocols.io, where a protocol can be assigned its own identifier (DOI) such that it can be cited independently in the future. For instructions see: http://journals.plos.org/plosone/s/submission-guidelines#loc-laboratory-protocols

We look forward to receiving your revised manuscript.

Kind regards,

Baochuan Lin, Ph.D.

Academic Editor

PLOS ONE

Journal Requirements:

Reviewers' comments:

Reviewer's Responses to Questions

**Comments to the Author**

1. Is the manuscript technically sound, and do the data support the conclusions?

Reviewer #1: No

Reviewer #2: No

2. Has the statistical analysis been performed appropriately and rigorously? 

Reviewer #1: Yes

Reviewer #2: No

3. Have the authors made all data underlying the findings in their manuscript fully available?

Reviewer #1: No

Reviewer #2: Yes

4. Is the manuscript presented in an intelligible fashion and written in standard English?

Reviewer #1: Yes

Reviewer #2: Yes

5. Review Comments to the Author

Reviewer #1: This is an important and interesting study on a newly emerging POC technology of TB diagnosis evaluated under actual condition. However, the paper is confused due to a poor study design. There are two serious concerns in this study that make the paper unacceptable.

1. The study aims to evaluate usefulness of a LAMP method in a practical setting in Thailand. The LAMP method is now available as an only commercial kit TB-LAMP assay (Loopamp™MTBC Detection Kit, Eiken Chemical Company Ltd., Japan) as endorsed by WHO in 2016. It seems that the method used in this study is a unique system at least partially. So, it is important to state explicitly that the target to be evaluated was an in-house LAMP and not one commercially available LAMP recommended by WHO.

2. In evaluating the sensitivity of the method, the authors used culture negative (clinically defined) cases, as well as bacteriologically confirmed cases, as a gold standard of the cases of TB. It may be difficult to admit the clinical diagnosis as a diagnostic basis for such a study as this, apart from clinical practice. Vice versa, the definition of the gold (conventional) standard for specificity (non-cases) should be reconsidered. The following paper may be of use in revising the paper; Kaku et al: Accuracy of LAMP-TB Method for Diagnosing Tuberculosis in Haiti. Jpn. J. Infect. Dis., 69, 488–492, 2016.

Reviewer #2: I think this paper has the potential to contribute more information to an ongoing area of research – that is, the performance of cheaper alternative diagnostic testing for TB in areas of high prevalence of disease. However, I cannot recommend publication of the results without major revision as I believe the diagnostic performance is misrepresented and the citations are misrepresenting the work of the other authors. I believe the authors just need to take greater care in summarizing the background to the current and in selecting the correct statistical methods. I’ve called out a few examples where the text mis-represents the citations, but a thorough re-look by the authors at all citations should be done.

Abstract/Background

“proven diagnostic performance” – this is both vague and too specific at the same time

“most of the results were validated” – the results aren’t validated, the assay is validated

The language surrounding people with possible TB needs to be updated throughout the paper - avoid the use of terms like "TB suspects" that increase the stigma surrounding this disease. http://www.stoptb.org/assets/documents/resources/publications/acsm/LanguageGuide_ForWeb20131110.pdf

The paper states repeatedly that there is little work published from resource-challenged settings, but this claim is not supported. Even the references given cite studies in such decentralized settings. Maybe it just hasn’t been done in Thailand? A better summary of the literature needs to be included. How does this compare to other studies? How is the TB LAMP test performed in this study compare to the TB LAMP tests in other published literature? A better focus on properly relating the current study to the body of work in the literature rather than trying to claim it is quite novel would actually strengthen the paper. There is merit in replication or demonstrating an important diagnostic in a new geographical area.

In-house vs commercialized kit is mentioned but not explained. And the position of this paper (what LAMP testing approach is used) is not properly placed in the context of what other papers are using and the potential impact on sensitivity/specificity.

The sensitivity/specificity of LAMP in other papers, settings, etc needs to be stated with numbers and not just alluded to. A proper, specific summary of the literature is lacking.

“In 2016, WHO suggested the use of LAMP assay for the diagnosis of pulmonary tuberculosis” – this is not quite right, WHO recommendations are very specific and it is important to get that right. From the abstract of the citation provided: “WHO recommends that TB-LAMP can be used as a replacement for microscopy for the diagnosis of pulmonary TB in adults with signs and symptoms of TB”. This needs to be stated correctly. Also, given the paper has mentioned in-house vs commercialized kits, it needs to be clarified that the WHO guidance refers only to the Eiken LAMP kit.

“LAMP assay has a low cost per test, does not required advanced technological facilities, and can be routinely practiced in general hospital laboratories [3].” Reference 3 doesn’t support this statement – it doesn’t say anywhere that the LAMP assay has a low cost per test. It says “Costs can be kept to a minimum if testing is limited to specimens from the most high-risk patients based on proper clinical assessments and national testing algorithms based on public health policies.” There are other publications on the cost of the LAMP assay for TB diagnosis. The authors might explain better the infrastructure/training needed for LAMP based on this reference and others.

Reference 5 doesn’t appear to really relate to the sentences it comes after. Reference 3 would make a lot more sense as it is a detailed overview of TB diagnostics including many molecular diagnostics.

Setting

The paper needs to do more to state what sets this setting apart from (or ties it to) other studies. See the methods section describing setting in reference 22 for how attributes of the specific site can be expressed in the context of the needs of LAMP.

Study Design

This is not a cross-sectional design, it is a prospective design. The plan was to prospectively enroll 120 patients.

“New patients who were clinically suspected of 109 pulmonary TB (coughing for more than two weeks with or without hemoptysis), aged more than 18 years old were consecutively invited into the study regardless of nation status.” Suggest re-writing to something more like: ‘Adults more than 18yrs of age with symptoms indicative of pulmonary TB (coughing…) and no history of TB were consecutively enrolled regardless of national status.’ If patients were ‘invited’ but not enrolled, we need numbers on how many declined.

“Samples with contaminated culture results or samples from patients who were previously documented as TB cases were excluded.” Were the patients excluded or the samples?

Methods

A map of which samples were used for what tests would be quite helpful. Highlight if any of the reference tests (smear, LJ culture, MGIT culture) were performed on the same sputum as LAMP.

Make it clear somewhere that smear-negative refers to AFB smear-negative.

Study size estimation

This has no purpose here – the study is done. Sample size estimation is for study planning purposes, for securing funding and making sure the plan has statistical validity.

Statistical analysis

The first four sentences are unnecessary.

The authors need to state what method was used to obtain the 95% CI for the sens/spec/PPV/NPV/LR+. It is clear from my testing that the Clopper Pearson binomial exact test was used, the authors should include the reference (usually found in the software documentation).

Kappa statistics are for inter-reader reliability, not for comparison of correlations between tests. It includes the concept that agreement may happen by chance when two people are guessing. However, it is not appropriate for comparison of diagnostic results because there isn’t guessing – the samples should not agree by chance but because they are or are not TB and the sensitivities of tests objectively vary. Spearman’s correlation can be used, but I think what you actually want is McNemar’s test. The desire is to compare the diagnostic performance (i.e. accuracy) between tests – McNemar’s test will do that. Alternatively Spearman’s correlation can look at the [objective] agreement between tests.

Results

Table 1 is dedicated to showing the patient clinical characteristics by culture status. The p-values shown test whether these characteristics differ significantly dependent on culture status. It is expected that gender, nationality, and age should not differ. Whereas it is also expected that chest x-rays and sputum quality would differ.

“The baseline demographic data between culture188 positive and negative patients were comparable except for the presence of cavitary lesions on 189 chest radiographs and the character of collected sputum (Table 1).”

Age, nationality, and gender are demographic data.

Chest x-ray and sputum quality are clinical characteristics.

Table 2 – re-check the NPV for parallel testing

There are a lot of LAMP-positive and AFB smear-positive patients with negative culture. Especially given that the tests are done on different sputum samples, these should be considered patients with probable TB and not used in assessing sensitivity and specificity.

There are too few smear-negative, culture-positive patients to assess sensitivity. Specificity should not be stratified by smear status, only sensitivity. For the reason above (that smear-positive, culture-negative patients shouldn’t be included in estimations of sensitivity/specificity of LAMP), what the paper is calling ‘smear-negative specificity’ should in fact be reported as the actual specificity of LAMP.

Table 2 – the p-values shown have no real meaning! If you want to compare accuracy of tests, you cannot do a p-value over the final accuracy measures among a bunch of tests. You need to compare tests 1 against another by using 2x2 grids and McNemar’s test. So, if you want to compare the accuracy of LAMP to the accuracy of AFB stain, you use the grid in Table 3 and McNemar’s test:

46 5

5 59 McNemar test p-value: 1.000

Discussion

“This study had demonstrated the pragmatic performance of the LAMP test, which was comparable to that of the conventional smear microscopy and the fluorescence microscopy.” Not true, the performance of LAMP as evaluated in this study was below that of smear microscopy.

“Although the sensitivity and specificity of the LAMP test were lower than that of the acid fast stain and the fluorescence stain, the comparative statistical test revealed non-significant results” This is still true when McNemar’s test is performed, but the right statistical tests need to be used in the paper. Furthermore, a non-significant result doesn't mean no difference, it means the difference is likely smaller than the power of the study to detect.

Put PPV/NPV in the context of the local prevalence of disease! State from the literature or reliable source what the prevalence of TB is in the hospital’s area of Thailand. I would suggest giving the readers an example: Given that prevalence and a group of 1000 patients, state how many would be true positives, false positive, true negatives, and false negatives. You can therefore assess what burden the different accuracies will place on the hospital. I.e. if the specificity is quite low and the sensitivity is higher, is that better? If the sensitivity is high and the specificity is lower, is that better? Relate this to the LR+.

“In the clinical context of TB diagnosis, both the LAMP test and the smear microscopy are considered as a diagnostic test which would normally be done in TB suspects with high pre-test probability [14]” – this is not what the reference says.

“Therefore, a serial test relying on both the result from the LAMP test and the acid-fast stain would be more appropriate for use as a rule-in test as it carried higher specificity and positive likelihood ratio than other methods.” Authors should define ‘rule-in’ test and what is generally expected of such a test. Should note the increased cost of such an approach.

The effect of a gold standard which is not itself perfect should be discussed. Also the variability between sputum samples should be discussed.

A better look at the differences between this study and others with better test performance needs to be done.

“Currently, the WHO only supported the use of two rapid molecular tests for the diagnosis of 294 pulmonary tuberculosis, which were Xpert MTB/RIF and the LAMP test” – as the concept of LAMP test from a kit and other LAMP tests has been raised, and the variability of accuracy depending, it needs to be clear that the WHO recommendation is only for the Eiken LAMP test kit!

6. PLOS authors have the option to publish the peer review history of their article (what does this mean?). If published, this will include your full peer review and any attached files.

Reviewer #1: Yes: Toru Mori

Reviewer #2: Yes: Christen M Gray

---

## [Author Response · Author response to Decision Letter 0]

7 Apr 2020

Responses to Reviewers’ comments

Pragmatic accuracy of loop-mediated isothermal amplification (LAMP)

for diagnosis of pulmonary tuberculosis in a Thai community hospital

Reviewer #1: 

1. The study aims to evaluate usefulness of a LAMP method in a practical setting in Thailand. The LAMP method is now available as an only commercial kit TB-LAMP assay (Loopamp™MTBC Detection Kit, Eiken Chemical Company Ltd., Japan) as endorsed by WHO in 2016. It seems that the method used in this study is a unique system at least partially. So, it is important to state explicitly that the target to be evaluated was an in-house LAMP and not one commercially available LAMP recommended by WHO.

o The LAMP test in our study was a non-commercial, in-house LAMP.

o We re-wrote the manuscript and emphasized that the test used was in-house LAMP.

2. In evaluating the sensitivity of the method, the authors used culture negative (clinically defined) cases, as well as bacteriologically confirmed cases, as a gold standard of the cases of TB. It may be difficult to admit the clinical diagnosis as a diagnostic basis for such a study as this, apart from clinical practice. Vice versa, the definition of the gold (conventional) standard for specificity (non-cases) should be reconsidered. The following paper may be of use in revising the paper; Kaku et al: Accuracy of LAMP-TB Method for Diagnosing Tuberculosis in Haiti. Jpn. J. Infect. Dis., 69, 488–492, 2016.

o We modified the inclusion criteria for analysis as suggested by both reviewers.

o As the analysis was done in a per-patient fashion, patients with smear-positive and culture-negative results would be excluded, as these patients were considered as probable TB cases. Therefore, the evaluation of sensitivity would include patients with both smear positive and smear negative with positive culture results. In contrast, the evaluation of specificity would include only patients with smear-negative and culture-negative results. 

 

Reviewer #2: 

1. Abstract/Background: “proven diagnostic performance” – this is both vague and too specific at the same time, “most of the results were validated” – the results aren’t validated, the assay is validated

o We rewrote the abstract and introduction part as suggested.

2. The language surrounding people with possible TB needs to be updated throughout the paper - avoid the use of terms like "TB suspects" that increase the stigma surrounding this disease. http://www.stoptb.org/assets/documents/resources/publications/acsm/LanguageGuide_ForWeb20131110.pdf

o We rewrote the abstract and introduction part as suggested.

3. The paper states repeatedly that there is little work published from resource-challenged settings, but this claim is not supported. Even the references given cite studies in such decentralized settings. Maybe it just hasn’t been done in Thailand? A better summary of the literature needs to be included. How does this compare to other studies? How is the TB LAMP test performed in this study compare to the TB LAMP tests in other published literature? A better focus on properly relating the current study to the body of work in the literature rather than trying to claim it is quite novel would actually strengthen the paper. There is merit in replication or demonstrating an important diagnostic in a new geographical area.

o We rewrote the abstract and introduction part as suggested.

4. In-house vs commercialized kit is mentioned but not explained. And the position of this paper (what LAMP testing approach is used) is not properly placed in the context of what other papers are using and the potential impact on sensitivity/specificity.

o We rewrote the abstract and introduction part as suggested.

5. The sensitivity/specificity of LAMP in other papers, settings, etc needs to be stated with numbers and not just alluded to. A proper, specific summary of the literature is lacking.

o We rewrote the abstract and introduction part as suggested.

6. “In 2016, WHO suggested the use of LAMP assay for the diagnosis of pulmonary tuberculosis” – this is not quite right, WHO recommendations are very specific and it is important to get that right. From the abstract of the citation provided: “WHO recommends that TB-LAMP can be used as a replacement for microscopy for the diagnosis of pulmonary TB in adults with signs and symptoms of TB”. This needs to be stated correctly. Also, given the paper has mentioned in-house vs commercialized kits, it needs to be clarified that the WHO guidance refers only to the Eiken LAMP kit.

o We rewrote the abstract and introduction part as suggested.

7. “LAMP assay has a low cost per test, does not required advanced technological facilities, and can be routinely practiced in general hospital laboratories [3].” Reference 3 doesn’t support this statement – it doesn’t say anywhere that the LAMP assay has a low cost per test. It says “Costs can be kept to a minimum if testing is limited to specimens from the most high-risk patients based on proper clinical assessments and national testing algorithms based on public health policies.” There are other publications on the cost of the LAMP assay for TB diagnosis. The authors might explain better the infrastructure/training needed for LAMP based on this reference and others.

o We rewrote the abstract and introduction part as suggested.

o We changed the references to the statement as follow: Sohn H. Cost, affordability, and cost-effectiveness of TB-LAMP assay. In: Report to WHO Guideline Development Group Meeting on TB-LAMP Assay. Edn. Geneva: World Health Organization; 2016 and Shete PB, Farr K, Strnad L, Gray CM, Cattamanchi A. Diagnostic accuracy of TB-LAMP for pulmonary tuberculosis: a systematic review and meta-analysis. BMC Infect Dis. 2019;19(1):268. Published 2019 Mar 19. doi:10.1186/s12879-019-3881-y

8. Reference 5 doesn’t appear to really relate to the sentences it comes after. Reference 3 would make a lot more sense as it is a detailed overview of TB diagnostics including many molecular diagnostics.

o We rewrote the abstract and introduction part as suggested.

Setting

1. The paper needs to do more to state what sets this setting apart from (or ties it to) other studies. See the methods section describing setting in reference 22 for how attributes of the specific site can be expressed in the context of the needs of LAMP.

o We elaborated the character of our setting as suggested:

o Level of health system: rural

o Distance to reference laboratory: 0 km 

o Median LAMP test workload: 6 (4-10)

o Electricity and backup power: infrequent power outages, power generator (350 Kw) and UPS (2.7 Kw)

o Biosafety cabinet infrastructure: BSC class II 

o Laboratory staff: 4 lab technicians, 1 lab assistant

2. Study Design: This is not a cross-sectional design; it is a prospective design. The plan was to prospectively enroll 120 patients.

o We changed the type of design to prospective diagnostic accuracy study as suggested.

o We would like to make a constructive argument on this point, as the diagnostic accuracy research is actually cross-sectional study in design. The cross-sectional design is only the type of membership condition, single component of study base, and cross-sectional design can therefore be collected prospectively or retrospectively. We would like to ask you to kindly refer to this reference: Assessment of the accuracy of diagnostic tests: the cross-sectional study by Knottnerus JA, 2003. 

Link: https://www.ncbi.nlm.nih.gov/pubmed/14615003

3. “New patients who were clinically suspected of 109 pulmonary TB (coughing for more than two weeks with or without hemoptysis), aged more than 18 years old were consecutively invited into the study regardless of nation status.” Suggest re-writing to something more like: ‘Adults more than 18yrs of age with symptoms indicative of pulmonary TB (coughing…) and no history of TB were consecutively enrolled regardless of national status.’ If patients were ‘invited’ but not enrolled, we need numbers on how many declined.

o We re-wrote the sentence as suggested: Adult patients aged more than 15 years old with symptoms indicative of pulmonary TB (coughing for more than two weeks with or without hemoptysis) and no history of TB were consecutively enrolled regardless of national status.

4. “Samples with contaminated culture results or samples from patients who were previously documented as TB cases were excluded.” Were the patients excluded or the samples?

o Patients with previously documented TB cases were excluded.

o Patients with two contaminated or missing culture results were excluded.

Methods

1. A map of which samples were used for what tests would be quite helpful. Highlight if any of the reference tests (smear, LJ culture, MGIT culture) were performed on the same sputum as LAMP.

o Conventional macroscopy, LAMP test, and culture were conducted as routinely done.

o All patients were given three sealed containers for the collection of morning sputum specimens. Of all containers sent to the laboratory, only the one with seemingly adequate sputum, containing both mucoid or mucopurulent characters with a sample volume more than 3 ml, was used for the whole investigation procedures as routinely done. Specimens were sent for smear microscopy with conventional acid-fast bacilli (AFB) staining with Ziehl-Neelsen technique and fluorescence acid-fast staining with Auramine O solution. 

2. Make it clear somewhere that smear-negative refers to AFB smear-negative.

o We added detail on the smear-negative status as suggested.

o According to WHO definitions, any patient with at least two AFB smears of scanty grade or one or more smears of 1+ or more was defined as smear-positive case. Smear-negative case was conversely defined.

3. Study size estimation

This has no purpose here – the study is done. Sample size estimation is for study planning purposes, for securing funding and making sure the plan has statistical validity.

o The study size estimation part was removed as suggested.

4. Statistical analysis. The first four sentences are unnecessary.

o The first four sentences were removed as suggested.

5. The authors need to state what method was used to obtain the 95% CI for the sens/spec/PPV/NPV/LR+. It is clear from my testing that the Clopper Pearson binomial exact test was used, the authors should include the reference (usually found in the software documentation).

o The 95% confidence intervals were calculated using the Clopper Pearson binomial exact method.

o We added this statement in the statistical section and added the citation as suggested.

6. Kappa statistics are for inter-reader reliability, not for comparison of correlations between tests. It includes the concept that agreement may happen by chance when two people are guessing. However, it is not appropriate for comparison of diagnostic results because there isn’t guessing – the samples should not agree by chance but because they are or are not TB and the sensitivities of tests objectively vary. Spearman’s correlation can be used, but I think what you actually want is McNemar’s test. The desire is to compare the diagnostic performance (i.e. accuracy) between tests – McNemar’s test will do that. Alternatively, Spearman’s correlation can look at the [objective] agreement between tests.

o Spearman’s rank correlation was inserted into the manuscript to represent the objective agreement between tests as suggested. 

o The agreement of LAMP test with smear microscopy methods was analyzed with Kappa’s statistics and Spearman’s rank correlation.

o We still presented the value of Kappa’s statistics as many of the previous studies on LAMP assay and other diagnostic tests had done [1–3].

 

Results

1. Table 1 is dedicated to showing the patient clinical characteristics by culture status. The p-values shown test whether these characteristics differ significantly dependent on culture status. It is expected that gender, nationality, and age should not differ. Whereas it is also expected that chest x-rays and sputum quality would differ. The baseline demographic data between culture188 positive and negative patients were comparable except for the presence of cavitary lesions on 189 chest radiographs and the character of collected sputum (Table 1). Age, nationality, and gender are demographic data. Chest x-ray and sputum quality are clinical characteristics.

o We reanalyzed all the data after exclusion of patients with probable TB (LAMP test positive and AFB smear positive patients with negative culture).

o All the baseline demographic and clinical characteristics data were reanalyzed and presented in Table 1.

o The statements in the results section were re-written as suggested.

2. Table 2 – re-check the NPV for parallel testing

o We reanalyzed all the data after exclusion of patients with probable TB (LAMP test positive and AFB smear positive patients with negative culture).

o All the data on Table 2 were checked for any error as suggested.

3. There are a lot of LAMP-positive and AFB smear-positive patients with negative culture. Especially given that the tests are done on different sputum samples, these should be considered patients with probable TB and not used in assessing sensitivity and specificity.

o We reanalyzed all the data after exclusion of patients with probable TB (LAMP test positive and AFB smear positive patients with negative culture).

o The final study size for analysis of LAMP test diagnostic accuracy was therefore 107 patients. (8 patients were excluded, 6 patients with both LAMP test and AFB smear-positive and culture negative, 1 patient with AFB positive and culture negative, and 1 patient with fluorescence stain positive and culture negative)

4. There are too few smear-negative, culture-positive patients to assess sensitivity. Specificity should not be stratified by smear status, only sensitivity. For the reason above (that smear-positive, culture-negative patients shouldn’t be included in estimations of sensitivity/specificity of LAMP), what the paper is calling ‘smear-negative specificity’ should in fact be reported as the actual specificity of LAMP.

o We exclude smear-positive, culture negative patients from the analysis as suggested.

o We reported the actual specificity of LAMP test without stratification.

o We acknowledged that our there are too few smear negative, culture positive patients to assess sensitivity in the discussion part.

5. Table 2 – the p-values shown have no real meaning! If you want to compare accuracy of tests, you cannot do a p-value over the final accuracy measures among a bunch of tests. You need to compare tests 1 against another by using 2x2 grids and McNemar’s test. So, if you want to compare the accuracy of LAMP to the accuracy of AFB stain, you use the grid in Table 3 and McNemar’s test:

o The comparison of diagnostic indices between LAMP test and AFB, fluorescence stain was re-analyzed using McNemar’s exact probability test as suggested. We presented the result of the pairwise tests separately and reformatted Table 2.

o Pairwise testing was not performed to compare the specificity between the LAMP test and the smear microscopy methods as the specificity of the latter was affected by incorporation bias and would not be comparable to the in-house LAMP.

o Table 3 was also reformatted.

o Spearman’s rank correlation was used as suggested. 

Discussion

1. “This study had demonstrated the pragmatic performance of the LAMP test, which was comparable to that of the conventional smear microscopy and the fluorescence microscopy.” Not true, the performance of LAMP as evaluated in this study was below that of smear microscopy.

o We rewrote the discussion part as suggested.

o “This study had demonstrated the pragmatic diagnostic performance of the in-house LAMP assay in a remote hospital of a high TB burden country. It was revealed that the overall sensitivity of the in-house LAMP in our study was lower than the numbers reported in the majority of the previous in-house LAMP studies. Nonetheless, the specificity was comparable to other figures reported in literature. In comparison to microscopy methods, the AFB and fluorescence stain, the in-house LAMP was found to be inferior in terms of overall sensitivity (82.0% vs. 88.0%, p=0.375) and accuracy (88.8% vs. 94.4%, p=1.000); however, the comparative statistical test revealed non-significant results. Based on the result of our study, we suggest that the in-house LAMP should not be a substitute to conventional smear methods, but should be done in parallel, which would result in a higher sensitivity with fewer false-negative TB cases.”

2. “Although the sensitivity and specificity of the LAMP test were lower than that of the acid-fast stain and the fluorescence stain, the comparative statistical test revealed non-significant results” This is still true when McNemar’s test is performed, but the right statistical tests need to be used in the paper. Furthermore, a non-significant result doesn't mean no difference, it means the difference is likely smaller than the power of the study to detect.

o We rewrote the discussion part as suggested.

o We reanalyzed our data using McNemar’s exact probability test as suggested.

3. Put PPV/NPV in the context of the local prevalence of disease! State from the literature or reliable source what the prevalence of TB is in the hospital’s area of Thailand. I would suggest giving the readers an example: Given that prevalence and a group of 1000 patients, state how many would be true positives, false positive, true negatives, and false negatives. You can therefore assess what burden the different accuracies will place on the hospital. I.e. if the specificity is quite low and the sensitivity is higher, is that better? If the sensitivity is high and the specificity is lower, is that better? Relate this to the LR+.

o We would like to make a constructive argument to this question as follow: The prevalence of culture-positive TB in this study was 46.7%. As this was a “consecutive recruitment of patients with sign and symptoms suggestive of pulmonary TB” or “patients with higher pre-test probability that the general prevalence” or the “person that the in-house LAMP test was intended to be used”, the calculation of positive predictive values could be directly calculated and reported from the study data as in the other study [1]. Moreover, both the in-house LAMP assay and acid-fast stain were not intended to be used as screening tests in the general population. For this reason, we did not include this part in our manuscript; however, we provide the answer to the question in this response paper.

o The latest Maesot’s population figures from the Health Data Center (HDC), the ministry of public health, Thailand, was 115,108 in 2019. The prevalence of pulmonary tuberculosis was 351 per 100,000 or 35 per 10,000.

 TB case Non-TB case Total 

LAMP positive 29 528 557 PPV 29/557=5.2%

LAMP negative 6 9,437 9,443 NPV 9437/9443=94.9%

Total 35 9,965 10,000 Prevalence=0.0035

4. “In the clinical context of TB diagnosis, both the LAMP test and the smear microscopy are considered as a diagnostic test which would normally be done in TB suspects with high pre-test probability [14]” – this is not what the reference says.

o The reference states “The TB LAMP assay is usually applied for TB-suspected patients and is rarely used for screening purpose. To rule-in the TB diagnosis, specificity is more important than sensitivity.”

o What we’re trying to imply from this statement was that the LAMP test was developed to be applied for patients who were suspicious of having TB with “higher pre-test probability than average person”. As the LAMP test was not for screening purpose, specificity is more important and should be more focused than sensitivity.

o After we re-analyzed the data with the exclusion of probable TB cases, our specificity increased to comparable level with previous studies. The parallel and serial testing was omitted from our analysis as the test accuracy of combination of the in-house LAMP with other smear microscopy methods would be seriously affected by incorporation bias (smear-positive, culture-negative patients were all excluded.

5. “Therefore, a serial test relying on both the result from the LAMP test and the acid-fast stain would be more appropriate for use as a rule-in test as it carried higher specificity and positive likelihood ratio than other methods.” Authors should define ‘rule-in’ test and what is generally expected of such a test. Should note the increased cost of such an approach.

o After we re-analyzed the data with the exclusion of probable TB cases, our specificity increased to comparable level with previous studies. The parallel and serial testing was omitted from our analysis as the test accuracy of combination of the in-house LAMP with other smear microscopy methods would be seriously affected by incorporation bias (smear-positive, culture-negative patients were all excluded.

6. The effect of a gold standard which is not itself perfect should be discussed. Also the variability between sputum samples should be discussed.

o The use of routine TB culture as a reference standard might be inadequate, as some TB patients could be classified as not having TB [6]. Different culture media and techniques could be used in composite to achieve different performance characteristics[4]. With a higher quality reference standard, the sensitivity of the in-house LAMP should be increased when a portion of three remaining false-positive cases was re-classified as true-positive cases.

o This study had a higher proportion of salivary sputum than mucous sputum. This could affect the diagnostic performance of both the index and the reference test[5]. The percentage of culture-positive TB cases was lower in salivary samples than in mucous samples (35.8% vs. 65.0%, p=0.005). Both the quality and quantity of sputum specimens were associated with positivity of smear, molecular testing methods (Xpert MTB/RIF and PCR), and TB culture [6,7]. Thus, it was possible that some patients with pulmonary TB might be classified as smear-negative, LAMP-negative, or even culture-negative cases. Interestingly, it was revealed from our data that the proportion of smear-positive, LAMP-positive results was also significantly lower in salivary sputum than in mucous sputum (31.3% vs 57.5%, p=0.009 and 29.9% vs. 60.0%, p=0.003, respectively). Therefore, the sensitivity and accuracy of all tests, including LAMP, might be underestimated. Previous studies reported that by improving the sputum quality, TB diagnostic yield increased[8,9]. Therefore, high-quality sputum collection must be encouraged both in practice and studies. 

7. A better look at the differences between this study and others with better test performance needs to be done.

o In this study, the sensitivity of the in-house LAMP test was 82.0% (95%CI 68.6-91.4) in culture-positive TB patients, respectively. In the past, several studies had reported a higher sensitivity of the in-house LAMP test, which ranges from 90.0 to 100.0%. Most of these studies were either University hospital, TB-specialized centers or hospitals, or national TB-specialized laboratory, which were generally equipped with highly-trained personnel and adequate infrastructural supports. The overall sensitivity of our in-house LAMP was consistent with two previous studies from India and Zambia, which was 79.5% (95%CI 64.0-89.0) and 81.4% (95%CI 71.6-89.0), respectively. Although both studies were performed in University hospitals, the LAMP procedures were modified to suit local conditions, and sputum processing and DNA extraction was done with commercial kits. The higher sensitivity of the acid-fast stain and the fluorescence stain in our study could be explained by the high prevalence of TB, the absence of HIV patient or a smaller number of patients with paucibacillary sputum, and the availability of skilled technicians

8. “Currently, the WHO only supported the use of two rapid molecular tests for the diagnosis of 294 pulmonary tuberculosis, which were Xpert MTB/RIF and the LAMP test” – as the concept of LAMP test from a kit and other LAMP tests has been raised, and the variability of accuracy depending, it needs to be clear that the WHO recommendation is only for the Eiken LAMP test kit!

o We edited the statement as follow: “Currently, the WHO only supported the use of two rapid molecular tests for the diagnosis of pulmonary tuberculosis, which were Xpert MTB/RIF and the commercialized TB-LAMP assay”.

 

References

1. George G, Mony P, Kenneth J. Comparison of the Efficacies of Loop-Mediated Isothermal Amplification, Fluorescence Smear Microscopy and Culture for the Diagnosis of Tuberculosis. PLoS ONE. 2011;6. doi:10.1371/journal.pone.0021007

2. Phetsuksiri B, Rudeeaneksin J, Srisungngam S, Bunchoo S, Klayut W, Nakajima C, et al. Comparison of Loop-Mediated Isothermal Amplification, Microscopy, Culture, and PCR for Diagnosis of Pulmonary Tuberculosis. Jpn J Infect Dis. 2020;advpub. doi:10.7883/yoken.JJID.2019.335

3. Wang Z, Sun H, Ren Z, Xue B, Lu J, Zhang H. Feasibility and Performance of Loop-Mediated Isothermal Amplification Assay in the Diagnosis of Pulmonary Tuberculosis in Decentralized Settings in Eastern China. BioMed Res Int. 2019;2019. doi:10.1155/2019/6845756

4. Cudahy P, Shenoi S. Diagnostics for pulmonary tuberculosis. Postgrad Med J. 2016;92: 187–193. doi:10.1136/postgradmedj-2015-133278

5. Shi J, Dong W, Ma Y, Liang Q, Shang Y, Wang F, et al. GeneXpert MTB/RIF Outperforms Mycobacterial Culture in Detecting Mycobacterium tuberculosis from Salivary Sputum. In: BioMed Research International [Internet]. 2018 [cited 12 Dec 2019]. Available: https://www.hindawi.com/journals/bmri/2018/1514381/

6. Ho J, Marks GB, Fox GJ. The impact of sputum quality on tuberculosis diagnosis: a systematic review. Int J Tuberc Lung Dis Off J Int Union Tuberc Lung Dis. 2015;19: 537–544. doi:10.5588/ijtld.14.0798

7. Yoon SH, Lee NK, Yim JJ. Impact of sputum gross appearance and volume on smear positivity of pulmonary tuberculosis: a prospective cohort study. BMC Infect Dis. 2012;12: 172. doi:10.1186/1471-2334-12-172

8. Hirooka T, Higuchi T, Tanaka N, Ogura T. [The value of proper sputum collection instruction in detection of acid-fast bacillus]. Kekkaku. 2004;79: 33–37. 

9. Sicsú AN, Salem JI, Fujimoto LBM, Gonzales RIC, Cardoso M do S de L, Palha PF. Educational intervention for collecting sputum for tuberculosis: a quasi-experimental study 1. Rev Lat Am Enfermagem. 2016;24. doi:10.1590/1518-8345.0363.2703

---

## [Decision Letter · Decision Letter 1]

13 May 2020

PONE-D-20-00432R1

Pragmatic accuracy of in-house loop-mediated isothermal amplification (LAMP) for diagnosis of pulmonary tuberculosis in a Thai community hospital

PLOS ONE

Dear Dr. Phinyo,

Thank you for submitting your manuscript to PLOS ONE. After careful consideration, we feel that it has merit but does not fully meet PLOS ONE’s publication criteria as it currently stands. Therefore, we invite you to submit a revised version of the manuscript that addresses the points raised during the review process.

Both reviewers agreed that the revised manuscript showed significant improvement, however, one of the reviewers still have concerns regarding clarity of the LAMP assay and discussion that need to be address carefully. Please see reviewer #2 insightful comments below.  In addition, the quality of the language needs to be improved to maintain the quality of published work, since PLoS ONE does not perform copyediting of manuscripts at any later stage in the publication process.  There are quite a few awkward sentences throughout the manuscript.  Please have a fluent, preferably native, English-language speaker thoroughly copyedit your manuscript for language usage, spelling, and grammar.  Personally, at a more detailed level, I found several issues need to be addressed, please see specific comments below.

Specific comments:

Abstract and Tables:  Change the format 95%CI to be consistent with the rest of the manuscript, for example“95%CI 78.3,97.5” should be “95%CI 78.3-97.5”.Line 80, implicated has negative meaning, suggest changing to implemented.Line 162, change M. *tuberculosis to **Mycobacterium*
*tuberculosis *since this is the first time you mentioned the bacteria*. *Also all the* “*M.” need to be italicized in M. *tuberculosis *throughout the manuscript*.*Line 192 – 193, should the “smear-positive and culture-positive results” be “smear-positive and culture-negative results”?Line 243 – 246:  These sentences need reference(s).

We would appreciate receiving your revised manuscript by Jun 27 2020 11:59PM. To enhance the reproducibility of your results, we recommend that if applicable you deposit your laboratory protocols in protocols.io, where a protocol can be assigned its own identifier (DOI) such that it can be cited independently in the future. For instructions see: http://journals.plos.org/plosone/s/submission-guidelines#loc-laboratory-protocols

We look forward to receiving your revised manuscript.

Kind regards,

Baochuan Lin, Ph.D.

Academic Editor

PLOS ONE

Reviewers' comments:

Reviewer's Responses to Questions

**Comments to the Author**

1. If the authors have adequately addressed your comments raised in a previous round of review and you feel that this manuscript is now acceptable for publication, you may indicate that here to bypass the “Comments to the Author” section, enter your conflict of interest statement in the “Confidential to Editor” section, and submit your "Accept" recommendation.

Reviewer #1: All comments have been addressed

Reviewer #2: (No Response)

2. Is the manuscript technically sound, and do the data support the conclusions?

Reviewer #1: Yes

Reviewer #2: Partly

3. Has the statistical analysis been performed appropriately and rigorously? 

Reviewer #1: Yes

Reviewer #2: Yes

4. Have the authors made all data underlying the findings in their manuscript fully available?

Reviewer #1: (No Response)

Reviewer #2: Yes

5. Is the manuscript presented in an intelligible fashion and written in standard English?

Reviewer #1: Yes

Reviewer #2: No

6. Review Comments to the Author

Reviewer #1: The reviewer is pleased to see that all of his comments are well addressed and the manuscript has been revised accordingly.

Reviewer #2: The English of the paper needs further work to be clear to the intended audience. While the main portion of the paper has been improved, further clarity around the assay (compared to other similar assays) is needed. Finally, the discussion requires heavy revision as it contains unsupported conclusions.

7. PLOS authors have the option to publish the peer review history of their article (what does this mean?). If published, this will include your full peer review and any attached files.

Reviewer #1: No

Reviewer #2: Yes: Christen M Gray

---

## [Author Response · Author response to Decision Letter 1]

14 May 2020

Response to Reviewers

We want to thank both the editor and the reviewers for granting the opportunity to revise our manuscript for publication in the PLOS one journal. We hope that our responses and revisions would substantially improve the quality of our manuscript and would be qualified for publication in the journal. If there were any further questions or minor points to be addressed or elaborated, please let us know. We would be more than eager to make any further revision. 

Editor’s comments

Specific comments:

1. Abstract and Tables: Change the format 95%CI to be consistent with the rest of the manuscript, for example “95%CI 78.3,97.5” should be “95%CI 78.3-97.5”.

• Changed as suggested.

2. Line 80, implicated has negative meaning, suggest changing to implemented.

• We modified the first two sentences as “As financial resources are usually limited in countries with high TB prevalence, a commercial TB-LAMP could still be unattainable. More affordable in-house LAMP assays were later developed and applied in several centers”.

3. Line 162 change M. tuberculosis to Mycobacterium tuberculosis since this is the first time you mentioned the bacteria. Also, all the “M.” need to be italicized in M. tuberculosis throughout the manuscript.

• Corrected as suggested.

4. Line 192 – 193, should the “smear-positive and culture-positive results” be “smear-positive and culture-negative results”?

• Corrected as suggested.

5. Line 243 – 246: These sentences need reference(s)

• We inserted some references to the two sentences as suggested. 

Reviewer’s comments

Thank you to the authors for the revisions made. This is a much better paper to present what is important work. However, I still have a few concerns. These focus on clarification of the ‘in-house assay’ and the discussion. Additionally, I think a review of the paper by a medical writer or any strong English editor would boost the communication of the results enormously.

1. The paper needs to be reviewed in detail for grammar and English. Other than general tidiness, in a number of places, the intent of what the authors are saying is lost due to odd grammar choices. For the best readability and better reach for the research contained, a review of the writing is recommended. I have made a few notes and suggestions in specific places.

a. We corrected all of your English suggestions.

b. We also modified and re-written some of the sentences in the manuscript to improve the readability.

2. The difference in assays still needs to be clearer. An ‘in-house assay’ is one that is not performed from a kit. You refer to ‘the in-house’ LAMP assay a lot as if there is only one, which is not the case. There are many papers out there with different ‘in-house’ LAMP assays. From the introduction, it sounds like you are presenting the findings from an in-house assay you developed following the protocol presented in Pandey et al. If so, this needs to be stated very clearly. However, from the methods section, it does not necessarily sound like you are not following that protocol and that this is a unique in-house assay. Please clarify in the paper.

a. We made the modification and improved the clarity of our in-house LAMP method as suggested.

3. When discussing previous results and meta-analyses, it needs to be clear that these refer to ‘in-house LAMP assays’ and not ‘the in-house LAMP assay’ as they are not uniform.

a. Corrected as suggested.

4. Inclusion of ‘Accuracy’ in Table 2 is a bit odd, but it can be kept if it is defined in the statistical methods section.

a. It was pre-specified in the methods section. 

5. The discussion has a lengthy discourse on the costs of Xpert vs LAMP. But there is no referencing of the studies that have costed these two in order to make a proper comparison. It feels quite unsupported.

a. We removed unsupported statements from the paragraph and make the paragraph more concise. 

6. In the discussion, the authors state ‘No previous study had officially addressed the effect of sputum quality on the LAMP test’. I’m not sure this is true and would caution the authors not to make such a sweeping statement. 

a. We removed the sentence out of the discussion section as suggested.

7. In the discussion, “Interestingly, it was revealed from our data that the proportion of smear-positive, LAMP-positive results was also significantly lower in salivary sputum than in mucous sputum (31.3% vs. 57.5%, p=0.009 and 29.9% vs. 60.0%, p=0.003, respectively). Therefore, the sensitivity and accuracy of all tests, including LAMP, might be underestimated.” 1) Do not present new results in the discussion – these need to be included in the Results section first. 3) are these sensivity? Specificity? Accuracy? 3) This is not an interpretation that makes sense. The sensitivity/specificity is reported based on the best sputum sample available from the patients – quality samples are difficult to obtain. You can instead interpret it as ‘Sensitivity and specificity would be improved if higher quality sputum is obtained’.

a. We modified the content as suggested.

b. We moved the findings to the results section.

8. In general, the discussion needs to be revised to make only statements supported by the literature, the study, or a comparison of the two. Much of the discussion feels like the authors musings.

a. We modified the whole discussion sections to be as objective as possible. 

9. In the discussion, I would suggest focusing on sensitivity and specificity and not accuracy as accuracy is not a common way of discussing or assessing diagnostic tests due to its difficulty of interpretation.

a. Corrected as suggested.

10. The references as displayed in the reference section aren’t quite right. In reference #1, instead of Lancet, the Journal is listed as Lancet Lond Engl which is not correct. This inclusion of a city occurs in reference #7 as well.

a. Corrected as suggested.

---

## [Decision Letter · Decision Letter 2]

9 Jun 2020

PONE-D-20-00432R2

Pragmatic accuracy of an in-house loop-mediated isothermal amplification (LAMP) for diagnosis of pulmonary tuberculosis in a Thai community hospital

PLOS ONE

Dear Dr. Phinyo,

Thank you for submitting your manuscript to PLOS ONE. After careful consideration, we feel that it has merit but does not fully meet PLOS ONE’s publication criteria as it currently stands. Therefore, we invite you to submit a revised version of the manuscript that addresses the points raised during the review process.

Specifically, the reviewer still has concerns that need to be addressed carefully. Please see reviewer's insightful comments below.

We look forward to receiving your revised manuscript.

Kind regards,

Baochuan Lin, Ph.D.

Academic Editor

PLOS ONE

Reviewers' comments:

Reviewer's Responses to Questions

**Comments to the Author**

1. If the authors have adequately addressed your comments raised in a previous round of review and you feel that this manuscript is now acceptable for publication, you may indicate that here to bypass the “Comments to the Author” section, enter your conflict of interest statement in the “Confidential to Editor” section, and submit your "Accept" recommendation.

Reviewer #2: All comments have been addressed

2. Is the manuscript technically sound, and do the data support the conclusions?

Reviewer #2: Yes

3. Has the statistical analysis been performed appropriately and rigorously? 

Reviewer #2: Yes

4. Have the authors made all data underlying the findings in their manuscript fully available?

Reviewer #2: Yes

5. Is the manuscript presented in an intelligible fashion and written in standard English?

Reviewer #2: Yes

6. Review Comments to the Author

Reviewer #2: The manuscript is much improved after the second revision. Some minor problems remain, but these are specific rather than general and should be able to be easily addressed.

- The use of ‘This’ at the beginning of sentences in the Background is not very clear.

- Some of the other corrections have also reduced clarity rather than enhanced, I have made a note for each instance for ease of author correction in the annoted pdf that I have attached to this review.

- Line 84, ‘More affordable in-house LAMP assays were later developed’ – this is still a problematic assertion as the first reference (the original reference for the sentence) is dated 2008 which is published prior to the availability and endorsement of the commercial TB-LAMP assay being compared. Also, none of the now 5 references compare affordability of the commercialized assay to an in-house assay to the commercial assay. Suggest the authors change to ‘An in-house LAMP assay may be more affordable…”

- Line 331, the authors state that no patients with HIV infection were included – was HIV testing performed on all patients? Were they confirmed to be HIV negative? Or should this say more simply that HIV testing was not performed?

- Line 341, authors should remove ‘Sensitivity and specificity would be improved if higher quality sputum is obtained’ – the authors have no way to confirm that. And, no support has been given as to why specificity would, even hypothetically, change.

- In the previous review, I noted that if the Kappa statistic 95% CI is displayed, then the Spearman Rho in the same table should also have the 95% CI. However, in retrospect, I realize that Spearman’s Rho should not have a 95% CI as one cannot be calculated for this method. Suggest removing and making a note.

7. PLOS authors have the option to publish the peer review history of their article (what does this mean?). If published, this will include your full peer review and any attached files.

Reviewer #2: Yes: Christen M Gray

---

## [Author Response · Author response to Decision Letter 2]

9 Jun 2020

Responses to Reviewers’ comments: minor revision II

Pragmatic accuracy of an in-house loop-mediated isothermal amplification (LAMP) for diagnosis of pulmonary tuberculosis in a Thai community hospital

Reviewer #2: The manuscript is much improved after the second revision. Some minor problems remain, but these are specific rather than general and should be able to be easily addressed.

- The use of ‘This’ at the beginning of sentences in the Background is not very clear.

Corrected as suggested

- These nucleic amplification techniques were known for yielding rapid and accurate TB diagnosis, which would overcome the limitations of classical methods (ie, insensitivity for smear microscopy and lengthy incubation period for TB culture).

- However, several obstacles remain for the application of these molecular tests as point-of-care testing in community settings due to their complexity to execute and substantial requirements for financial and personnel resources.

- Some of the other corrections have also reduced clarity rather than enhanced, I have made a note for each instance for ease of author correction in the annoted pdf that I have attached to this review.

- Corrected as suggested in the reviewer’s attached pdf file.

- Line 84, ‘More affordable in-house LAMP assays were later developed’ – this is still a problematic assertion as the first reference (the original reference for the sentence) is dated 2008 which is published prior to the availability and endorsement of the commercial TB-LAMP assay being compared. Also, none of the now 5 references compare affordability of the commercialized assay to an in-house assay to the commercial assay. Suggest the authors change to ‘An in-house LAMP assay may be more affordable…”

- Corrected as suggested.

- An in-house LAMP assay may be more affordable to the commercial one [11–15].

- Line 331, the authors state that no patients with HIV infection were included – was HIV testing performed on all patients? Were they confirmed to be HIV negative? Or should this say more simply that HIV testing was not performed?

- All patients with symptoms suggestive of TB were offered routine HIV counselling and HIV rapid antibody tests. (We added this within the methods section)

- All included patients had negative HIV results. (we added this within the results section)

- Line 341, authors should remove ‘Sensitivity and specificity would be improved if higher quality sputum is obtained’ – the authors have no way to confirm that. And, no support has been given as to why specificity would, even hypothetically, change.

- We removed the sentence as suggested.

In the previous review, I noted that if the Kappa statistic 95% CI is displayed, then the Spearman Rho in the same table should also have the 95% CI. However, in retrospect, I realize that Spearman’s Rho should not have a 95% CI as one cannot be calculated for this method. Suggest removing and making a note.

- We removed the 95%CI as suggested and added a note under the table that the 95% confidence interval of the spearman’s rank correlation is not estimable.

---

## [Decision Letter · Decision Letter 3]

9 Jul 2020

Pragmatic accuracy of an in-house loop-mediated isothermal amplification (LAMP) for diagnosis of pulmonary tuberculosis in a Thai community hospital

PONE-D-20-00432R3

Dear Dr. Phinyo,

We’re pleased to inform you that your manuscript has been judged scientifically suitable for publication and will be formally accepted for publication once it meets all outstanding technical requirements.  Please spell out M. tuberculosis on line 153 since this is the first time the name of the bacteria was mentioned.

Kind regards,

Baochuan Lin, Ph.D.

Academic Editor

PLOS ONE

Additional Editor Comments (optional):

Reviewers' comments:

Reviewer's Responses to Questions

**Comments to the Author**

1. If the authors have adequately addressed your comments raised in a previous round of review and you feel that this manuscript is now acceptable for publication, you may indicate that here to bypass the “Comments to the Author” section, enter your conflict of interest statement in the “Confidential to Editor” section, and submit your "Accept" recommendation.

Reviewer #2: All comments have been addressed

2. Is the manuscript technically sound, and do the data support the conclusions?

Reviewer #2: Yes

3. Has the statistical analysis been performed appropriately and rigorously? 

Reviewer #2: Yes

4. Have the authors made all data underlying the findings in their manuscript fully available?

Reviewer #2: Yes

5. Is the manuscript presented in an intelligible fashion and written in standard English?

Reviewer #2: Yes

6. Review Comments to the Author

Reviewer #2: The authors have done much to improve the manuscript and have addressed all of my concerns.

One very tiny thing: M. tuberculosis should be fully written out the first time it appears in the text and the abbreviation given. This occurs on the 2nd mention.

7. PLOS authors have the option to publish the peer review history of their article (what does this mean?). If published, this will include your full peer review and any attached files.

Reviewer #2: **Yes: **Christen M Gray

---

## [Editor Report · Acceptance letter]

10 Jul 2020

PONE-D-20-00432R3 

Pragmatic accuracy of an in-house loop-mediated isothermal amplification (LAMP) for diagnosis of pulmonary tuberculosis in a Thai community hospital 

Dear Dr. Phinyo:

I'm pleased to inform you that your manuscript has been deemed suitable for publication in PLOS ONE. Congratulations! Your manuscript is now with our production department. 

Kind regards, 

on behalf of

Dr. Baochuan Lin 

Academic Editor

PLOS ONE